# Post COVID-19 Syndrome in Patients with Asymptomatic/Mild Form

**DOI:** 10.3390/pathogens10111408

**Published:** 2021-10-30

**Authors:** Annа Malkova, Igor Kudryavtsev, Anna Starshinova, Dmitry Kudlay, Yulia Zinchenko, Anzhela Glushkova, Piotr Yablonskiy, Yehuda Shoenfeld

**Affiliations:** 1Medical Department, St-Petersburg State University, 199034 Saint-Petersburg, Russia; piotr_yablonskii@mail.ru (P.Y.); yehuda.shoenfeld@sheba.health.gov.il (Y.S.); 2Department of Immunology, Institution of Experimental Medicine, 197376 Saint-Petersburg, Russia; igorek1981@yandex.ru; 3Almazov National Medical Research Centre, 197341 Saint-Petersburg, Russia; starshinova_aa@almazovcentre.ru; 4Medical Department, I.M. Sechenov First Moscow State Medical University, 119435 Moscow, Russia; D624254@gmail.com; 5NRC Institute of Immunology FMBA of Russia, 115478 Moscow, Russia; 6St. Petersburg Research Institute of Phthisiopulmonology, 199034 Saint-Petersburg, Russia; ulia-zinchenko@yandex.ru; 7V.M. Bekhterev National Research Medical Center for Psychiatry and Neurology, 192019 Saint-Petersburg, Russia; angela_glushkova@yahoo.com; 8Ariel University, Kiryat HaMada 3, Ariel 40700, Israel; 9Zabludowicz Center for Autoimmune Diseases, Sheba Medical Center, Tel-Hashomer 5265601, Israel

**Keywords:** COVID-19, asymptomatic, mild, Post COVID-19 Syndrome, autoimmune, anosmia, fatigue, Myalgic Encephalomyelitis/Chronic Fatigue Syndrome, autoimmune dysautonomia

## Abstract

Post COVID-19 Syndrome (PCS) is a complex of various symptoms developing a month or more after the acute phase of the disease. The cases of PCS development among patients with asymptomatic/mild forms are frequently reported; however, the pathogenesis of PCS in this group of patients is still not completely clear. The publications about COVID-19 which were published in online databases from December 2019 to September 2021 are analyzed in this review. According to the analysis, PCS develops on average in 30–60% of patients, mainly among women. Fatigue, shortness of breath, cough, and anosmia were reported as the most common symptoms. The possible association between the described PCS symptoms and brain damage was revealed. We assume the possibility of an alternative course of COVID-19, which develops in genetically predisposed individuals with a stronger immune response, in which it predominantly affects the cells of the nervous system, possibly with the presence of an autoimmune component, which might have similarity with chronic fatigue syndrome or autoimmune disautonomia. Thus, the gender (female) and the presence of anosmia during an asymptomatic or mild course of the disease can be predictive factors for the development of PCS, which can be caused by autoimmune damage to neurons, glia, and cerebral vessels.

## 1. Introduction

The new coronavirus infection was first reported in December 2019, and it has claimed the lives of people around the world. Currently, there concern is growing about the consequences after suffering COVID-19 caused by SARS-CoV-2 virus (severe acute respiratory syndrome-related coronavirus 2).

SARS-CoV-2 enters into cells that only express ACE2 (angiotensin-converting enzyme 2) but not into cells without ACE2 or cells expressing other coronavirus receptors such as aminopeptidase N and dipeptidyl peptidase 4 (DPP4), suggesting that ACE2 is a unique cellular receptor for SARS-CoV-2 [1]. Extensive organ damage occurs because ACE2 is highly expressed not only for cells of the lung and vascular endothelium, but also on myocardial cells, renal proximal tubule and bladder urothelial cells, and cells of the testis, intestine, and liver [2]. Moreover, the presence of a long-term damaging viral agent, hyper inflammatory reactions, and a decrease in the expression of ACE 2 downregulate recovery processes [3,4].

The complication of the disease, including mild course, can be manifested by the development of the so-called Post COVID-19 Syndrome (PCS), which does not have clear criteria for its prevention, diagnosis, and management tactics. At the moment, there is no single approach to its definition and classification. Various authors classify complications depending on the genesis, time of its onset, and the form of acute infection (Table 1), for example, the consequences of complications during the illness, complications of treatment, or reemerging symptoms (Table 1).

The most common symptoms were fatigue (35–72%), dyspnea (29–65%), sleep disturbance (57%), cough (43%), asthenia (40%), memory problems (34%), anosmia (21–23%), and arthralgia (20–22%) [8]. According to various databases, 20% of patients had at least one symptom of PCS during the month or more after the disease, and 10% of patients during 3 or more months after [9,10]. According to the British National Institute for Health Research, PCS developed in a significant proportion of non-hospitalized patients; at least one symptom was observed after a month among 20–30% of patients, and after 3 months among 10% [11].

Given the widespread infection and the high level of PCS development even in asymptomatic and mild forms, it is necessary to reveal the pathogenic features of the syndrome development in these patients to identify high-risk groups.

## 2. Methods

The review analyzes publications about COVID-19 in online databases “Medline”/“PubMed” and “Scopus” from December 2019 to September 2021. The first selection of articles was based on the keywords COVID-19, asymptomatic, mild, pathogenesis, immune system, Post COVID-19 Syndrome, and autoimmune, anosmia. The inclusion criteria were asymptomatic or mild forms of COVID-19 diagnosed with PCR test, the development of PCS symptoms a minimum of 1 month after, and the description of the developed PCS symptoms. Due to the small number of published papers and the variability of samples studied, we considered it impossible to carry out a systematic review of high quality following the recent PRISMA guidelines.

## 3. Results

At this moment there are a few studies of patients with asymptomatic or mild forms of COVID-19 (Table 2).

Analysis of the described data allows us to do several conclusions:PCS developed among 30–60% of patients with asymptomatic or mild forms of COVID-19 on average.The most common symptoms were fatigue, shortness of breath, cough, anosmia, and ageusia. Headaches, brain fog, and other symptoms of central nervous system damage were also reported.Most common PCS occurs among women (on average 60%).

## 4. Discussion

### 4.1. Pathogenesis of the Described Complications

The cause of multiorgan damage due to coronavirus infection may be explained by the direct effect on cells by viruses as well as immune-mediated and vascular complications [23]. However, there is no consensus on the pathogenesis of the development of the particular symptoms currently.

#### 4.1.1. Fatigue

According to Nauen et al.’s hypothesis, fatigue may be caused by the dysfunction of the capillaries of the brain [24]. Perhaps this could explain the development of postural orthostatic tachycardia syndrome (POTS) after SARS-CoV-2 infection, which is defined by heart rate increase > 30 beats per minute after head-up tilting test [25], palpitations, and tachycardia without obvious cardiac abnormalities [26]. This autonomic dysfunction, including orthostatic hypotension, vasovagal syncope, and POTS, was described in patients with mild forms of COVID-19 [27]. The autoimmune origin of these diseases is considered [28,29] with autoantibodies to α-/β-adrenoceptors and muscarinic receptors detected [30].

According to Townsend et al. [31], patients with fatigue that developed an average of 166 days after infection did not have the autonomic dysfunction criteria from Ewing’s battery parameters. The researchers found an association between fatigue and anxiety that was not diagnosed in any participant before COVID-19. However, the researchers did not perform autoantibodies detection; therefore, it is impossible to exclude autoimmune etiology of developed fatigue.

Association between fatigue and anxiety suggests the presence of Myalgic Encephalomyelitis/Chronic Fatigue Syndrome (ME/CFS) [32]. ME/CFS is a chronic disease with autoimmune origin [33] characterized by unexplained fatigue after exercise, and symptoms are associated with cognitive, immunological, endocrinological, and autonomic dysfunction, which are currently attributed to brain damage [34]. The main target of ME/CFS is considered to be the brain stress center, a cluster of neurons in the paraventricular nucleus of the hypothalamus [35,36], which might be damaged by autoantibodies to 5-hydroxytryptamine, gangliosides, and phospholipids [33]. It is assumed that infection with SARS-CoV-2 in certain individuals may be a trigger for the development of ME/CFS, being a serious physiological stressor. Thus, in a study of a small group of patients, according to the diagnostic criteria, CFS was detected 6 months after infection [37]. It is important to note that ME/CFS is more common among women [38], and symptoms of fatigue and anxiety are common after coronavirus infection.

#### 4.1.2. Shortness of Breath, Cough

According to the Liam Townsend study, the majority of patients with mild forms had a normal chest X-ray with a decrease in desaturation during the six-minute walking test (6MWT) 75 days after diagnosis [39]; therefore, fibrosis is not the cause of shortness breath in mild forms.

According to the hypothesis of Townsend et al. [40], delayed lung damage after SARS-CoV-2 infection may be caused by an autoimmune response to ACE2, which can be explained by the forced presentation of the ACE2 protein in complex with CoV spike in Fc receptor-positive antigen-presenting cells. One of the explanations of the development of autoimmune complications might be also the molecular similarity of SARS-CoV-2 S-protein with surfactant proteins, which was shown in the Kanduc et al. study [41]. According to Cappello F et al., hypoxia during the infection might induce expression of antistress proteins, including heat shock proteins (Hsp), that have a high level of structural conservation and share various similar antigens within and across species. It might be the cause of immunological crossreactivity between Hsp/chaperones and virus proteins that leads to autoimmune reactivity [42]. Generated antibodies might damage endothelium cells with the development of thrombosis, disseminated intravascular coagulation, and multi-organ failure [43,44].

The possible cross-reactions leading to the activation of autoimmune processes in the lungs damaging pneumocytes and endotheliocytes can be considered. These findings might allow to assume the presence of the delayed interstitial pneumonia and fibrosis after a longer period; therefore, during the clinical studies, these inflammatory autoimmune changes are not so well expressed for X-ray imaging.

#### 4.1.3. Anosmia

The cause of anosmia is considered to be the dysfunction of lymphatic drainage from the periventricular organs and the direct viral invasion into the extracellular spaces of the olfactory epithelium [45,46]. Significant amounts of SARS-CoV S protein and RNA have been found in the cells of the olfactory mucosa [47]. According to Gordon et al., one of the mechanisms of penetration of the virus into the cells of the olfactory epithelium may be interaction with opioid receptors sigma-1 and sigma-2 [48]. It is worth noting that anosmia is also more common among female patients [49].

#### 4.1.4. Headaches, Brain, and Other CNS Symptoms

The development of headaches and other CNS symptoms may be the result of virus damage to both brain cells and blood vessels, which was shown on autopsy materials (inflammatory foci, changes in the parenchyma and blood vessels) [50,51]. It is assumed that prolonged inflammation, accompanied by the release of TNF-α, can cause cognitive-behavioral changes [52,53].

The evidence of the presence of viral RNA in the brain and cerebrospinal fluid (CSF) has been published; however, according to the researchers, it does not indicate a true infection with SARS-CoV-2 [54]. On the other hand, ACE2 expression was found in neuronal and glial cells of the central nervous system [55], in the olfactory mucosa [56]. Meinhardt et al. suggest that the virus can enter through the olfactory tract into certain neuroanatomical areas, including the main control center of the respiratory and cardiovascular systems in the medulla oblongata [47], which might explain the development of shortness of breath in patients.

### 4.2. The Hypothesis

According to the analysis of the literature, it can be assumed that the most common symptoms of PCS are characterized by the presence of nerve cell damage and have similarities with CFS and disautonomia, which are autoimmune diseases. According to the study by Wallukat et al. [57], patients with the symptoms mostly of neurological origin (including post-COVID-19 fatigue, attention deficit, tremor and others) were positive for antibodies to chronotropic GPCR-fAABs targeting the β2-adrenoceptor, the α1-adrenoceptor, the angiotensin II AT1-receptor, and the nociception-like opioid receptor and negative to chronotropic GPCR-fAABs targeting the muscarinic M2-receptor, the MAS-receptor, and the ETA-receptor, which are also found in POTS and dysautonomia [58]. In patients who have had a mild and severe coronavirus infection, there are significant changes in the composition of circulating immune cells, which persist for a long time (at least 3 to 9 months after the acute phase of the disease) and are similar to autoimmune diseases.

Firstly, this is an increase in the proportion of pro-inflammatory T-helper 17 (Th17) cells circulating in the blood for a long time after the disease and a decrease in anti-inflammatory T-regulatory cells (Tregs), accompanied by a change in the Th17/Treg balance [59,60,61], which is specifically for a wide range of autoimmune diseases [62].

Secondly, an imbalance between T-follicular helper (Tfh) cell subpopulations and regulatory (Tfr) was observed [63], which is also noted in autoimmune manifestations associated with the formation of autoantibodies [64,65].

Thirdly, in the peripheral blood of COVID-19 patients, an increased number of short-living, highly differentiated CD8^+^ T-lymphocytes remaining for a long time was found [58,59,60,61,64], which indicates the ongoing processes of differentiation and circulation of these cells from the lymphoid tissue. These cells, apparently, are not associated more with the elimination of SARS-CoV-2 and its antigens, but may be another sign of the development of an autoimmune reaction [66]. An increase in the proportion of mature effector cells with the CD45RA^+^CCR7^−^ phenotype in the peripheral blood of convalescent patients was noted, and in the case of CD8^+^ cytotoxic T cells, this was also associated with an increase in the proportion of mature perforin- and granzyme-expressing lymphocytes, which are capable of producing TNFα and IFNγ [59]. Moreover, the presence of pulmonary complications was closely associated with an increase in the circulation of the proportion of short-living effector CD27–CD62L–CD8^+^ T cells, as well as mature T cells capable of producing perforin, granzyme B, and IFNγ [60]. It was also noted that 1.5–2.5 months after the acute phase of COVID-19, a high level of granzyme^+^CD8^+^ T cells remained in the peripheral blood of the recovered patients [67].

An important feature in the development of PCS after the asymptomatic/mild form is the predominance of female patients [14,20,68]. It is known that the female gender is a protective factor against the development of severe infections [69]. This is facilitated by several physiological characteristics, in particular:-immunomodulatory functions of estrogen and immunosuppressive functions of testosterone [70,71];-higher expression and poly-morphisms of ACE-2 and TMPRSS2 genes associated with a genetic predisposition to COVID-19 in men [72,73,74];-higher number of activated and terminally differentiated T cell populations (CD38 and HLA-DR-positive activated T cells) among women [75];-higher average serum concentrations of SARS-CoV-2 IgG antibodies at an early stage of infection among women [76,77].

With a more pronounced immune response, women are more predisposed to developing autoimmune diseases [78], in particular ME/CFS, fibromyalgia, etc.

We assume that among patients with a stronger immune response, the virus, when it enters the upper respiratory tract, damages the cells of the olfactory mucosa predominantly, which develops anosmia, and then is transferred to the brain by lymphatic drainage or axonal transport. After a little while, an inflammatory reaction with an autoimmune component develops, which might explain the development of CNS damage symptoms (fatigue, headaches, etc.) after a month or more. The virus can also damage the respiratory and cardiovascular centers, which might explain shortness of breath.

According to the hypothesis of Wostyn, the development of anosmia might be a predictive factor of the development of PCS, in particular fatigue [79], which might confirm the concept of the proposed pathogenesis (Figure 1).

We assume that there is an alternative form of the course of coronavirus infection, mainly affecting the cells of the nervous system and developing in genetically predisposed individuals (in particular among women). One of the genetic predisposing factors, besides the expression of ACE 2 and the protein associated with it, can be genotypes HLA-A*02:01 and HLA-A*03:01, associated with a low probability of developing a severe form of COVID-19 [80], and HLA-DRB1*04: 01 associated with asymptomatic infection [81] (Figure 2).

It is worth considering other complications of coronavirus infection in patients with mild forms of the disease, mediated by thrombotic or endocrine complications. Joseph et al. [82] described four clinical cases of thromboebolism in patients with a mild form. However, in all cases, complications occurred earlier than 2 weeks after the first symptoms, which does not fit into the PCS and indicates a second wave of organ damage by the virus in a more severe form [83]. A similar conclusion could be done from the clinical case of a long-segment arterial cerebral vessel thrombosis after mild form of COVID-19 14 days after the first symptoms, described by Sartoretti et al. [84].

## 5. Conclusions

According to the analysis of original articles on PCS among asymptomatic/mild COVID-19 patients, PCS develops on average in 30–60% of patients, mainly among women, with fatigue, shortness of breath, cough, and anosmia being the most common symptoms. The possible association between the described PCS symptoms and brain damage during coronavirus infection suggests an alternative form of the course of the disease that develops in genetically predisposed individuals with a stronger immune response (in particular women), in which it predominantly affects the cells of the nervous system with the presence of an autoimmune component, which has similarities with Myalgic Encephalomyelitis/Chronic Fatigue Syndrome or autoimmune disautonomia. In summary, the female gender and the presence of anosmia during an asymptomatic or mild form of the disease could be predictive factors of the development of PCS, which might be caused by autoimmune damage to neurons, glia, and cerebral vessels.

## Figures and Tables

**Figure 1 pathogens-10-01408-f001:**
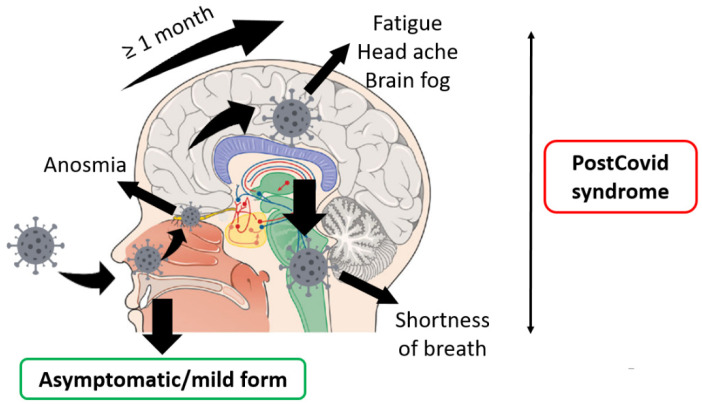
Schematic explanation of the PCS pathogenesis affecting CNS.

**Figure 2 pathogens-10-01408-f002:**
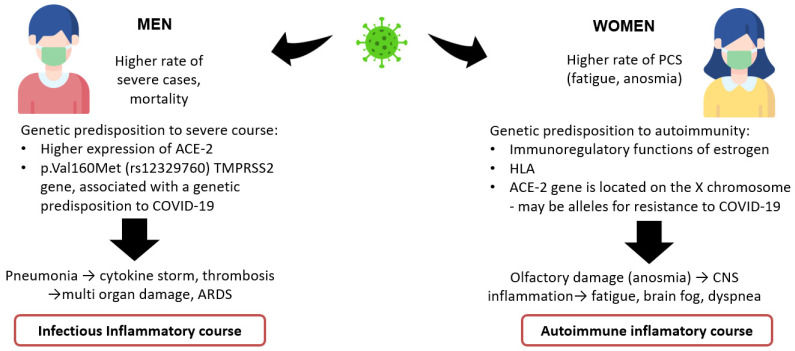
Hypothesis on different forms of COVID-19.

**Table 1 pathogens-10-01408-t001:** Proposed PCS classifications.

Amenta EM et al. [5]	Ceravolo MG et al. [6]	Centers for Disease Control and Prevention [7]
residual symptoms that persist after recovery from acute SARS-CoV-2 infection	symptoms persisting from the acute phase and their treatment	persistent COVID series of symptoms last for weeks or months
symptoms due to single or multiple organ dysfunction that persists after initial recovery	symptoms associated with a new disease	symptoms resulting from damage to multiple organs, such as the heart, lung, kidneys, skin, and nervous system
novel symptoms or syndromes that arise after mild or asymptomatic infection	late-onset symptoms resulting from COVID-19 arising at the end of the acute phase	consequences of COVID-19 treatment or prolonged hospitalisation
	impact on a previous pathology or disability	

**Table 2 pathogens-10-01408-t002:** The data of the studies on PCS among patients with asymptomatic or mild forms of COVID-19.

Authors	Sample	Period of Observation	Symptoms (%)	PCS (%)
Augustin M et al. [12]	958 SARS-CoV-2-convalescent patients, the majority initially presented with absent to mild symptoms	4 months	anosmia (12.4), ageusia (11.1), fatigue (9.7), and shortness of breath (8.6)	27.8
		7 months		34.8
Tenforde MW et al. [13]	292 young patients (mean age: 42.5 years) with mild COVID-19 16 days after diagnosis	2–3 weeks	cough, fatigue, and dyspnea	35.0
Carvalho-Schneider C et al. [14]	150 patients with mild COVID–19	2 months	asthenia (40.0), dyspnoea (30.0) anosmia/ageusia (23.0)	66.7
Yong Huang et al. [15]	1407 records No hospitalized	60 days	shortness of breath, chest pain, cough, or abdominal pain	27.0
Tabacof L et al. [16]	84 less severe acute infection with PCS	151 (54 to 255) days	fatigue (92.0%), loss of concentration/memory (74.0), weakness (68.0), headache (65.0), and dizziness (64.0)	All 84 were with PCS
Logue JK et al. [17]	177 11 (6.2%) were asymptomatic, 150 (84.7%) were outpatients with mild illness, and 16 (9.0%) had moderate or severe disease requiring hospitalization	9 months	fatigue (13.6) and loss of sense of smell or taste (13.6). 23 patients (13.0%) reported other symptoms, including brain fog (2.3)	32.7
Melanie L. Bell et al. [18]	303 no hospitalized	30–59 days post-diagnosis	fatigue (37.5), shortness of breath (37.5), brain fog (30.8), and stress/anxiety (30.8)	68.7
		≥60 days		73.0
Havervall S et al. [19]	323 (94%) seropositive	2 months 8 months	anosmia, fatigue, ageusia, and dyspnea	26.0 15.0
	and 1072 (84%) seronegative participants	2 months 8 months		9.0 3.0
Bliddal S et al. [20]	445 Danish non-hospitalized Completely asymptomatic COVID-19 was reported by 34%	≥4 weeks	fatigue (16.0), concentration or memory difficulties (13.0), reduced sense of smell (10.0), and shortness of breath (10.0)	36.0
		≥12 weeks	fatigue (16.0) and concentration difficulties (13.0)	40.0
Fernández-de-Las-Peñas C et al. [21]	9011 non-hospitalized patients	≥one post-COVID-19 symptom at 30, 60, or ≥90days after onset	Fatigue and dyspnea 35.0–60.0 cough (20.0–25.0), anosmia (10.0–20.0), ageusia (15.0–20.0), or joint pain (15.0–20.0)	45.9
Andrews PJ et al. [22]	Mild to moderate Health Care Workers 114	52 days	olfactory 73.1 taste alteration 69.2	

## Data Availability

All data generated or analyzed during this study are included in this published article.

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
