# Peer review of "Post COVID-19 Syndrome in Patients with Asymptomatic/Mild Form"

_pathogens, 2021, doi:10.3390/pathogens10111408_

Round 1
Reviewer 1 Report
In my opinion, this manuscript is well written and its contribution is very relevant for the field. In general, I have not major comments on this work. I would only clarify if it is possible to provide more details about literature search, study selection etc. Is it possible to carry out a systematic review? Is it possible to follow the PRISMA guidelines?
Minor change:
Please change "sample" with population in Table 2
Author Response
In my opinion, this manuscript is well written and its contribution is very relevant for the field. In general, I have not major comments on this work.
I would only clarify if it is possible to provide more details about literature search, study selection etc. Is it possible to carry out a systematic review? Is it possible to follow the PRISMA guidelines?
The answer: the paragraph “Methods” was added
Minor change:
Please change "sample" with population in Table 2
The answer: was corrected
Reviewer 2 Report
This is an interesting paper about post COVID-19 syndrome in patients with asymptomatic/mild form. I particularly appreciated the hint to autoimmunity. However, Authors can not ignore the hypothesis of molecular mimicry as the "primum movens" for generating autoimmunity. I suggest them to read the papers of my and other groups on this topic and I'd like very much if they can add this piece of information to their paper (e.g., in the paragraph "3.2. The hypothesis").
Here some references, but they will be able to find also others:
1: Cappello F. Is COVID-19 a proteiform disease inducing also molecular mimicry phenomena? Cell Stress Chaperones. 2020 May;25(3):381-382. doi: 10.1007/s12192-020-01112-1. Epub 2020 Apr 20. PMID: 32314313; PMCID: PMC7167495.
2: Cappello F, Gammazza AM, Dieli F, de Macario, Macario AJ. Does SARS-CoV-2 Trigger Stress-InducedAutoimmunity by Molecular Mimicry? A Hypothesis. J Clin Med. 2020 Jun 29;9(7):2038. doi: 10.3390/jcm9072038. PMID: 32610587; PMCID: PMC7408943.
3: Marino Gammazza A, Légaré S, Lo Bosco G, Fucarino A, Angileri F, Conway de Macario E, Macario AJ, Cappello F. Human molecular chaperones share with SARS- CoV-2 antigenic epitopes potentially capable of eliciting autoimmunity against endothelial cells: possible role of molecular mimicry in COVID-19. Cell Stress Chaperones. 2020 Sep;25(5):737-741. doi: 10.1007/s12192-020-01148-3. Epub 2020 Aug 4. PMID: 32754823; PMCID: PMC7402394.
As minor issues: please note that the manuscript lacks of a paragraph about Methods (actually, it has been briefly described at the end of the first paragraph, Introduction); please note that paragraph 5 (Conclusions) should be renumbered as paragraph 4.
Author Response
This is an interesting paper about post COVID-19 syndrome in patients with asymptomatic/mild form. I particularly appreciated the hint to autoimmunity.
However, Authors can not ignore the hypothesis of molecular mimicry as the "primum movens" for generating autoimmunity. I suggest them to read the papers of my and other groups on this topic and I'd like very much if they can add this piece of information to their paper (e.g., in the paragraph "3.2. The hypothesis").
Here some references, but they will be able to find also others:
1: Cappello F. Is COVID-19 a proteiform disease inducing also molecular mimicry phenomena? Cell Stress Chaperones. 2020 May;25(3):381-382. doi: 10.1007/s12192-020-01112-1. Epub 2020 Apr 20. PMID: 32314313; PMCID: PMC7167495.
2: Cappello F, Gammazza AM, Dieli F, de Macario, Macario AJ. Does SARS-CoV-2 Trigger Stress-InducedAutoimmunity by Molecular Mimicry? A Hypothesis. J Clin Med. 2020 Jun 29;9(7):2038. doi: 10.3390/jcm9072038. PMID: 32610587; PMCID: PMC7408943.
3: Marino Gammazza A, Légaré S, Lo Bosco G, Fucarino A, Angileri F, Conway de Macario E, Macario AJ, Cappello F. Human molecular chaperones share with SARS- CoV-2 antigenic epitopes potentially capable of eliciting autoimmunity against endothelial cells: possible role of molecular mimicry in COVID-19. Cell Stress Chaperones. 2020 Sep;25(5):737-741. doi: 10.1007/s12192-020-01148-3. Epub 2020 Aug 4. PMID: 32754823; PMCID: PMC7402394.
The answer: Thank you very much for suggested papers! The following part was included:
According to Cappello F et al hypoxia during the infection might induce expression of antistress proteins, including heat shock proteins (Hsp), that have high degree of structural conservation and share various similar antigens within and across species. It might be the cause of immunological crossreactivity between Hsp/chaperones and virus proteins which leads to autoimmune reactivity (Cappello F, Gammazza AM, Dieli F, de Macario, Macario AJ. Does SARS-CoV-2 Trigger Stress-InducedAutoimmunity by Molecular Mimicry? A Hypothesis. J Clin Med. 2020;9(7):2038. Published 2020 Jun 29. doi:10.3390/jcm9072038). Generated antibodies might damage endothelium cells with the development of thrombosis, disseminated intravascular coagulation and multi-organ failure (Cappello F. Is COVID-19 a proteiform disease inducing also molecular mimicry phenomena?. Cell Stress Chaperones. 2020;25(3):381-382. doi:10.1007/s12192-020-01112-1; Marino Gammazza A, Légaré S, Lo Bosco G, Fucarino A, Angileri F, Conway de Macario E, Macario AJ, Cappello F. Human molecular chaperones share with SARS- CoV-2 antigenic epitopes potentially capable of eliciting autoimmunity against endothelial cells: possible role of molecular mimicry in COVID-19. Cell Stress Chaperones. 2020 Sep;25(5):737-741. doi: 10.1007/s12192-020-01148-3. Epub 2020 Aug 4. PMID: 32754823; PMCID: PMC7402394).
As minor issues: please note that the manuscript lacks of a paragraph about Methods (actually, it has been briefly described at the end of the first paragraph, Introduction);
The answer: the paragraph “Methods” was added
please note that paragraph 5 (Conclusions) should be renumbered as paragraph 4.
Round 2
Reviewer 2 Report
The Authors modified the paper accordingly to my suggestions.